materials science

poly(N,N-dimethylaminoethyl methacrylate), antimicrobial properties, polymerization

**Author for correspondence:**
Dawid Stawski
e-mail: dawid.stawski@p.lodz.pl

This article has been edited by the Royal Society of Chemistry, including the commissioning, peer review process and editorial aspects up to the point of acceptance.

# Antibacterial properties of poly (*N,N*-dimethylaminoethyl methacrylate) obtained at different initiator concentrations in solution polymerization

Dawid Stawski[1], Karolina Rolińska[1], Dorota Zielińska[1,2], Priyanka Sahariah[3], Martha Á. Hjálmarsdóttir[4] and Már Másson[3]

[1]Institute of Material Technologies of Textiles and Polymer Composites, Lodz University of Technology, Lodz, Poland
[2]R&D Project Department, Institute of Security Technologies 'MORATEX', Lodz, Poland
[3]Faculty of Pharmaceutical Sciences, School of Health Sciences, University of Iceland, Reykjavík, Iceland
[4]Faculty of Medicine, Department of Biomedical Science, University of Iceland, Stapi, Hringbraut 31,101 Reykjavík, Iceland

DS, 0000-0002-7916-0239; KR, 0000-0002-4357-0260; DZ, 0000-0002-9691-7467

The samples of poly(*N,N*-dimethylaminoethyl methacrylate) were synthesized by radical polymerization. The amount of monomer and solvent was constant as opposed to an amount of initiator which was changing. No clear relationship between polymerization conditions and the molecular weight of the polymer was found, probably due to the branched configuration of produced polymer. Bactericidal interactions in all samples against Gram-positive and Gram-negative bacteria have been demonstrated. However, the observed effect has various intensities, depending on the type of bacteria and the type of sample.

## 1. Introduction

Polyelectrolytes are a sort of macromolecular compound that have ionogenic groups embedded in their structure, and that allow for electrolytic dissociation. Side groups of this kind can give the macromolecules specific properties, that non-electrolytic

polymers lack. For example, an introduction of a sufficiently large amount of hydrophilic groups makes the linear polymer become soluble in water.

The stability of dissociable binding in polyelectrolytes depends on the structure of the system and on the conditions in which it is located. In solvents with a high dielectric constant, the ionic system dissociates electrolytically into oppositely charged ions, the sum of which equals zero. The dissociation rate of polyelectrolytes depends mainly on the type of the ionic group and the solvent, as well as on other conditions of conducting the process and can reach values from almost zero to one. According to the definition, strong electrolytes dissociate to a much greater extent than weak ones. The dissociated ions create an electric field of high potential around themselves. On the other hand, the undissociated system, externally neutral electrically, generates a vestigial field derived from an ion-binding dipole, in which the charges are shifted to one of the system components. In polyelectrolyte with a high degree of dissociation, the charge of the entire macromolecule can be very large due to the presence of many groups with a strong charge of the same type. Changing the external conditions, especially the change in pH, can cause significant changes in the charge of the polyelectrolyte macromolecule, due to the change in the value of the electric field around it.

The very high concentration of ionogenic groups along the polyelectrolyte macromolecule, small distances between functional groups, and generated strong electrostatic field cause a clear effect on the material systems located in a sufficiently small distance. As a result, changes in the behaviour of foreign ions found in the immediate vicinity of a given macromolecule may be observed, inter alia. The impact of the electrostatic field rapidly decreases as the distance from the polyelectrolyte macromolecule increases, it is also highly deformed because its conformational structure usually takes the form of a glomus and is dynamically variable over time.

Poly(N,N-dimethylaminoethyl methacrylate) (PDMAEMA) (structure 1) is a linear polyelectrolyte with an amino group in the side chain, where the ester bond is located. It is a temperature-responsive polymer and at higher temperatures, the polymer phase gets separated from the solution as a result of breakage of hydrogen bonds [1–3].

structure 1

PDMAEMA easy dissolves in aqueous solution but its solubility is pH-dependent. PDMAEMA at pH 7 is partially hydrophobic, partially hydrophilic, exhibiting ampholytic nature. At low pH, the polymer is positively charged, at a value greater than 7 it becomes uncharged [4]. PDAMEMA draws the researchers' attention for a long time mostly because it proves excellent antimicrobial properties [5–8]. Antimicrobial cationic polymers feature chemical stability and non-volatility, presenting long-term activity. Poly(N,N-dimethylaminoethyl methacrylate) is a mucoadhesive antimicrobial polymer that can be quaternized with alkylating agents [9,10].

PDAMEMA is a polymer obtained by conventional radical polymerization. However, it should be remembered that it is difficult to polymerize. Several techniques have been developed in the literature to obtain a polymer with varying degrees of polymerization and polydispersity.

The most commonly used polymerization methods are listed in table 1.

Finding the relationship between average molecular weight and the level of antibacterial properties is an important issue. Low molecular analogues often do not have bioactive properties, which means that this parameter is related to the degree of polymerization. In our previous works, we have shown very interesting, nonlinear relationships of this type for chitosan derivatives [19,20]. The purpose of this work is to determine the possible occurrence of such dependence for PDMEAMA.

Microbial pathogens like the yeast of genera Candida, bacteria of genera Burkholderia, Pseudomonas, Salmonella, Serratia, Yersinia, Enterococcus, Escherichia, Staphylococcus or Streptococcus, and the fungus Cryptococcus neoformans have long been a threat to human health and social development [21]. The control and prevention of microbial infections has become a daunting challenge, for many years.

Lee-Anne B. Rawlinson et al. [6] showed that the antibacterial effect was dependent on the bacteria type. PDMAEMA is bacteriostatic against Gram-negative bacteria with MIC values between 0.1 and 1 mg cm$^{-13}$ but MIC values against Gram-positive bacteria were variable.

**Table 1.** Preparation of linear poly (*N*,*N*-dimethylaminoethyl methacrylate).

| reference | solvent | initiator | temperature [°C] | time [h] | Mn | Mw/Mn |
|---|---|---|---|---|---|---|
| [11] | tetrahydrofuran | AIBN | 65 | 24 | 113 000 | 1,21 |
| [12] | no solvent | 2,2-dimethoxy-2-phenyl acetophenone (photoinitiator) | | 30 min | | |
| [13] | water | ammonium persulfate | 60 | 24 | | |
| [14] | ethanol | | | | | |
| [6] | toluene | | 70 | >15 min | | |
| [5] | no solvent | | | | 1500–35 100 | |
| [15] | xylene | AIBN | 70 | | 63 000 | |
| [16] | DMF, Isoprop. alcohol, Ethanol | AIBN | 60 | 24 h | 9000–150 000 | |
| [17] | | AIBN | 60 | 48 h | | |
| [18] | 1 M HCl in water | ammonium peroxodisulfate | 60 | 22 h | 131 000 | 6.24 |
| | toluene | AIBN | 60 | 22 h | 20 000 | 3.15 |
| | | | | | 75 000 | 4.12 |

Fang Yao *et al.* [8] prepared microporous polypropylene hollow fibre membranes with surface-grafted block copolymer brushes of poly(ethylene glycol) monomethacrylate and 2-(dimethylamino)ethyl methacrylate. After quaternization of the tertiary amine groups of the PDMAEMA block, the membrane imparts the effective antibacterial and anti-fouling properties. They have found that the surface-functionalized membranes exhibit long-term durability and antibacterial efficacy in repeated applications. They suggested that such a modification of membranes can extend the potential applications as biomedical materials for blood oxygenation and haemodialysis.

Jinyu Huang *et al.* [5] modified polypropylene surface by coating PDMAEMA. The tertiary amine groups in PDMAEMA were converted to quaternary ammonium salts. They have found that the biocidal activity of the resultant surfaces depends on the amount of the grafted polymers. With the same grafting density, the surface grafted with molecular weight greater than $10\,000$ g mol$^{-1}$ showed almost 100% killing efficiency whereas a low biocidal activity (85%) was observed for the surface grafted with shorter chains (Mn = 1500 g mol$^{-1}$).

It was shown that PDMAEMA has antimicrobial properties but PDMAEMA has also been used to transfect cells. According to the literature, PDMAEMA has shown promising transfection activity [18,22–31]. It can be attached to plasmid DNA by electrostatic interactions [18]. The authors showed the relationship between molecular weight, polymer/plasmid ratio, the composition of the polymer and transfection efficiency [18]. Low pH and low ionic strength favour the formation of relatively small polyplexes [22]. Stawski *et al.* successfully deposited PDMAEMA on textiles [7]. They show that samples with external PDMAEMA layers are excellently active against *Staphylococcus aureus* under dynamic contact conditions. By contrast, samples finished with deposited silver showed in that case little antimicrobial effect. In one paper, a poly(*N*,*N*-dimethylaminoethyl methacrylate) nonwoven was produced by the blowing out technique [32]. It was found that the obtained nonwoven has antibacterial properties against *Staphylococcus aureus* and *Escherichia coli*.

Inspired by the potential interest of PDMAEMA as a medical compound the present study aims to verify antimicrobial properties of PDMAEMA with different molecular weight.

# 2. Material and methods

## 2.1. Materials

— *N*,*N*-dimethylaminoethyl methacrylate (DMAEMA) (Sigma-Aldrich, Germany) was purified by distillation under a vacuum (69–70°C, 1–2 mmHg).

**Table 2.** Polymerization conditions of DMAEMA.

| sample number | initiator (mg) | monomer (cm$^3$) | solvent (cm$^3$) |
|---|---|---|---|
| 1 | 0.114 | 25 | 52 |
| 2 | 0.104 | | |
| 3 | 0.094 | | |
| 4 | 0.084 | | |
| 5 | 0.074 | | |

— *p*-Xylene for synthesis (Sigma-Aldrich, Germany) was used without any further purification.
— Azobisisobutyronitrile (AIBN), (Merck, Germany) was used without any further purification.
— *n*-heptane (Sigma-Aldrich, Germany) was used without any further purification.

## 2.2. Methods

PDMAEMA was prepared by radical, solvent polymerization of DMAEMA initiated with azobisisobutyronitrile in xylene [32] according to contents shown in table 2. The purified DMAEMA was mixed with AIBN and xylene and maintained in a reflux condenser at a boil for 2 h. After cooling, the product was precipitated with *n*-heptane and dried under vacuum at room temperature.

## 2.3. FTIR spectroscopy

The PDMAEMA was characterized using a Thermo Scientific Nicolet spectrophotometer with KBr pellets. The spectra were obtained from 4000 to 400 cm$^{-1}$. Approximately 0.1–1.0% sample was well mixed into 200–250 mg fine potassium bromide powder and then finely pulverized and put into a pellet-forming die. A force of approximately 7 tons was applied under a vacuum of several mm Hg for about 5 min to form transparent pellets. Degassing was performed to eliminate air and moisture from the KBr powder.

Before forming the KBr powder into pellets, salt was drying at 110°C overnight. After drying the powder, KBr was stored in a desiccator. The background was measured with an empty pellet holder inserted into the sample chamber.

## 2.4. Determination of average molecular weight

The average molecular weight of synthesized polymers was measured by used two methods:

### 2.4.1. By gel permeation chromatography

Molecular weight (Mw) determination was carried out using gel permeation chromatography (GPC) following a previously published procedure [33]. GPC measurements were done using the Polymer Standards Service (PSS) (GmbH, Mainz, Germany), Dionex Ultimate 3000 HPLC system (Thermo Scientific-Dionex Softron GmbH, Germering, Germany), Dionex Ultimate 3000 HPLC pump, and Dionex Ultimate 3000 autosampler (Thermo Scientific-Dionex Softron GmbH, Germering, Germany), Shodex RI-101 refractive index detector (Shodex/Showa Denko Europe GmbH, Munich, Germany) and PSS's ETA-2010 viscometer. WINGPC Unity 7.4 software (PSS GmbH, Mainz, Germany) was used for data collection and processing. A series of three columns [Novema 10 µ guard (50 × 8 mm), Novema 10 µ 30 Å (150 × 8 mm) and Novema 10 µ 1000 Å (300 × 8 mm)] (PSS GmbH, Mainz, Germany) were used in the HPLC system. Poly(2-vinylpyridine) standards with Mp (1310–256000 Da) and Dextran (Mw = 62 400 Da), from PSS (GmbH, Mainz, Germany) were used for calibration. The eluent used was 0.1 M NaCl/0.1% TFA solution. Each sample was dissolved in the same eluent as mentioned above, at a concentration of 2 mg cm$^{-13}$, filtered through a 0.45 cm$^3$ filter (Spartan 13/0.45 RC, Whatman) before measurement at 25°C using a flow rate of 1 cm$^3$/min. Each sample had an injection volume of 100 cm$^3$ and a retention time of 30 min and all the measurements were done in triplicate.

**Table 3.** Parameters used to determine $k$ and $\alpha$ constants for PDMAEMA in a solution of 0.001 M NaOH in methanol at $T = 23.5°C$.

| $[\eta]$ | M | logM | log$[\eta]$ |
|---|---|---|---|
| 0.13 | 33 500 | 4.53 | −0.89 |
| 0.19 | 47 000 | 4.67 | −0.72 |
| 0.38 | 143 000 | 5.16 | −0.42 |
| 0.64 | 151 000 | 5.18 | −0.19 |
| 0.76 | 196 000 | 5.29 | −0.12 |
| 1.12 | 337 000 | 5.53 | 0.05 |
| 2.08 | 403 000 | 5.61 | 0.32 |
| 0.96 | 422 000 | 5.63 | −0.02 |
| 1.12 | 437 000 | 5.64 | 0.05 |

### 2.4.2. By viscometric method

Measurement of the viscometric average molecular weight was made using the Ubbelohde suspended-level viscometer. Samples were dissolved in a solution of 0.001 M NaOH in methanol. The tests were performed at the temperature of 23.5°C. All solutions were filtered through suitable Schott filters to remove microgels and impurities. The $k$ and $\alpha$ constants for the Mark–Houwink equation were previously determined by Prof. Połowiński in the Department of Physical Chemistry Polymers of the Lodz University of Technology for a given polymer-solvent-temperature system, using the fractionation method [34]. Briefly, for the samples after fractionation with the distribution close to the monomolecular one, the limit viscosity number and mean molecular weight were determined by the chromatographic method. The obtained values are shown in table 3.

Based on the data of table 2, a graph of dependence logM = f(log$[\eta]$) was made, that met the linearity of the function at the correlation coefficient $R^2 = 0.919$. The k and $\alpha$ parameters for the Mark–Houwink equation were determined from the graph in such a way that it obtained the following form:

$$[\eta] = 1.092 \times 10^{-5} \times M^{0.9}. \tag{2.1}$$

## 2.5. Determination of minimum inhibitory concentration and minimum lethal concentration

The antibacterial tests were assayed according to standard CLSI methods for antimicrobial dilution susceptibility tests [35]. Minimum inhibitory concentration (MIC) and minimum lethal concentration (MLC) values were measured [33] against two species of Gram-positive bacteria *Staphylococcus aureus* (*S. au*reus, ATCC 29213) and *Enterococcus faecalis* (*E. faecalis*, ATCC 29212) and two of Gram-negative bacteria *Escherichia coli* (*E. coli*, ATCC 25922) and *Pseudomonas aeruginosa* (*P. aeruginosa*, ATCC 27853) obtained from the American Type Culture Collection, representing strains that are susceptible to routinely measured antibiotics. The broth microdilution method was used to determine the MIC values using Mueller-Hinton Broth (Oxoid, Hampshire, UK) at pH 7.2 as the medium. Blood agar (heart infusion agar (Oxoid) with 5% (v/v) defibrinated horse blood) was used for the measurement of MLC. The samples were prepared by dissolving sample in sterile water to an initial concentration of 32.768 µg cm$^{-13}$. Fifty microliters of each sample were added to the first two wells on a micro-titer plate and twofold dilutions were done in 50 cm$^3$ of Mueller-Hinton broth from well two onwards. That gave a final range varying from 16.384 µg cm$^{-13}$ to 16 µg cm$^{-13}$. Gentamicin was used as the performance control during the test. A standard 0.5 McFarland suspension ($1–2 \times 10^8$ CFU cm$^{-3}$) was prepared by direct colony suspension in Mueller-Hinton broth. This suspension was then further diluted 100-fold so as to achieve a final test concentration of bacteria of approximately $1 \times 10^6$ CFU cm$^{-3}$ (or final concentration of $5 \times 10^5$ CFU/well in the microtiter plate). The microtiter plates were then incubated at 35°C for 18 h under moistened conditions. The MIC values were determined as the lowest concentration of the antibacterial agent that completely inhibited the visible growth of the microorganism in microtiter wells. For MLC measurement, 10 µl × 2 of each of the dilutions that showed no visible growth was plated on a blood agar plate and incubated at 35°C for 18 h. MLC was determined as the lowest concentration that achieved a 99.9% decrease in viable cells.

**Table 4.** Average molecular weights of obtained polymers determined by GPC.

| sample number | average molecular weight (Mw) [kDa] determined by GPC | polydispersity (D) as a GPC result | average molecular weight (Mv) [kDa] determined by viscometry |
|---|---|---|---|
| 1 | 55.54 | 1.49 | 64.77 |
| 2 | 238.17 | 3.80 | 107.16 |
| 3 | 50.85 | 1.63 | 86.34 |
| 4 | 57.35 | 1.61 | 69.30 |
| 5 | 63.49 | 1.80 | 76.52 |

## 2.6. Measurement of kill curves

The bactericidal effect for polymer 1 was determined using the time-kill assay based on the CLSI (NCCLS) protocol for the measurement of time-dependent bactericidal activities of antibacterial agents [36,37]. The bacteria used were the same strains of *S. aureus*, *E. faecalis*, *E. coli* and *P. aeruginosa* that were used for MIC/MLC determinations. The test concentration was $2 \times \mathrm{MIC}$ and the time points for viable cell count were 0, 0.5, 1, 2, 3, 4 and 8 h. Standard 0.5 McFarland bacterial suspension was prepared and added to the tubes containing broth and the test concentration of the antibacterial agent to get a final bacterial concentration of approximately $5 \times 10^5$ CFU cm$^{-3}$. The tubes were then incubated at 37°C and 100 µl samples were taken out at different time intervals. At each time point, 100 µl of the samples were serially diluted 1 : 10 in saline, 10 µl of the undiluted, and the diluted samples were plated in duplicate on Mueller Hinton agar and incubated at 37°C overnight. The dilutions that showed 20–200 colonies were then counted and the number of colonies for each sample was determined by averaging the obtained counts and CFU cm$^{-3}$ was calculated. The difference in the number of colonies ($\Delta$log10CFU cm$^3$) for the control and the antibacterial agent versus time intervals was then plotted.

# 3. Results and discussion

One of the basic parameters that determine the properties of polymers is their molecular weight. All the advantages of polymers that distinguish them from low molecular weight compounds are related to their extensive chain structure. In the radical polymerization of the reaction product, the basic parameter to condition the degree of polymerization is the amount of initiator. With a constant amount of monomer and unchanging other parameters of the process, such as temperature, time and solvent (in solution polymerization), it is the number of active centres generated by the disintegrating initiator that determines how many macromolecules the monomer becomes split into. During this research, the molecular weight was controlled by using variable amounts of the initiator. Based on viscosity tests and gel permeation chromatography, the molecular weights of the obtained polymers were determined, as shown in table 4.

The obtained values of the average molecular weights of the synthesized polymers are unexpected. Polymerization of monomers with ionogenic groups is more difficult and more complicated than that of neutral monomers, which is why suitable functional groups are often not introduced until the final polymer. In the case of DMAEMA, tertiary amine groups are present in the monomer, so they can impact the process mechanism.

FTIR and NMR spectroscopy were used as a tool for the analysis of obtained samples. All spectra look very similar. In figure 1, you can see one of these spectra (made for Sample 1), with characteristic signals marked on it. The peak between 2700–3000 cm$^{-1}$ is connected with the C−H bond from –N(CH$_3$)$_2$ groups. The carbonyl signal from ester groups appears between 1600 and 1800 cm$^{-1}$. Deformation vibrations from methylene groups on the main chain appear at 1400–1500 cm$^{-1}$ and signals connected with C−N bond on the side chain are at 1150 and 750 cm$^{-1}$.

The macromolecules of linear polymers in the solid body usually occur in the form of a glomus with the ability to rotate around the bonds between carbon atoms. Such a state is the most stable one because it is characterized by the lowest energy level and the highest value of entropy. In a solution, under the solvating action of the solvent, the linear polymer molecules take a more or less upright form, while the undissociated polyelectrolyte macromolecule has the shape of a cumulous, ball-like particle. If the

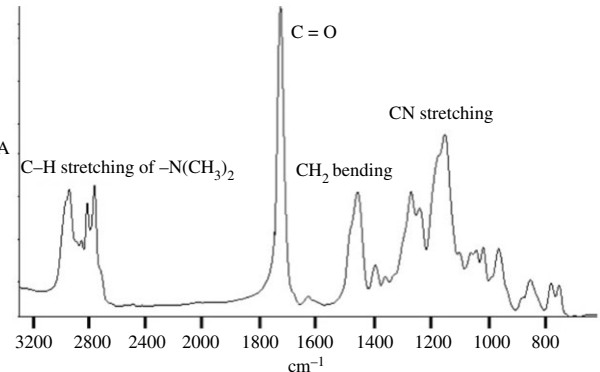

**Figure 1.** FTIR spectrum of Sample 1.

conditions existing in the solution change, causing an increase in the degree of dissociation of ionogenic groups, then the phenomenon of Coulomb repulsion of the same charges will occur, which will cause partial inflation of a glomus and increase of its volume; in the case of complete dissociation, the macromolecule takes on a wand shape. The dependence of the polyelectrolyte structure on the degree of dissociation, and hence on the external conditions existing in the solution, makes it difficult to determine their average molecular masses by viscometry methods. For the shape of a glomus, Einstein's law for colloidal solutions is rather applicable, where there is virtually no dependence of viscosity on the diameter. However, when a macromolecule has taken the shape of a fully straightened wand, the viscosity of the solutions can be described by the Mark-Houwink equation. Thus, changing the shape of the macromolecule at the same molecular weight causes deviations from the above-mentioned rights. By analysing the character of these deviations, one can therefore infer the shape of molecules in solution.

For the PMAEMA samples obtained in this research, there is no clear dependence of the values of the molecular weight averages on the amount of the initiator used. This kind of conclusion can be drawn from both viscosity and chromatographic measurements. This is a surprising observation, which can be explained by the formation of macromolecules with branched configuration under the conditions of the solution polymerization process carried out in the present work. With gel permeation chromatography, the delayed elution of branched macromolecules occurs due to the anchoring of loose branches in the pores of the bed and between the grains, which results in adulteration of the obtained results. Similarly, with the viscosity methods, the solution flow times are extended along with the increase of the degree of branching, resulting in unrepresentative results.

The five polymers were assayed for determining the antibacterial effect towards two Gram-positive (*S. aureus, E. faecalis*) and two Gram-negative (*E. coli* and *P. aeruginosa*) strains. The MIC values for the polymers showed that they were in general slightly more active towards the Gram-negative susceptible *E. coli* strain, while they showed the lowest activity towards highly resistant Gram-negative strain *P. aeruginosa*. Their activity towards the Gram positive *S. aureus* and *E. faecalis* was virtually the same, but somewhat stronger towards *E. coli* (table 5). No significant differences in the inhibitory concentrations of the five compounds were observed towards all the strains except *P. aeruginosa*, where Sample 2 showed the highest activity with MIC of 512 µg cm$^{-13}$ and polymers 3 and 5 were the least active ones (MIC = 4096 µg cm$^{-13}$). However, the differences in the susceptibility of the strains could be observed in the MLC values of the polymers. The lethal concentrations varied from 1024 to 2048 µg cm$^{-13}$ against *S. aureus*, 512–1024 µg cm$^{-13}$ against *E. faecalis*, 128–256 µg cm$^{-13}$ against *E. coli* and 512–4096 µg cm$^{-13}$ against *P. aeruginosa*, which suggests that the polymers have a better bactericidal effect towards the Gram-negative strains as compared to the Gram-positive strains.

To further evaluate the bactericidal effect of the polymers, the time-kill curve was determined for the four different strains in the presence of the most active compound (Sample 2) and compared to the number of viable cells in broth without any sample at the selected time points. The difference in the bacterial count in the presence and absence of Sample 2 with time is presented in figure 2.

A plot of the viable bacterial count in the growth control was also done and it shows that the count increased rapidly up to 9 logs within 0.5 h, followed by a slight increase up to 4 h, and finally it remained constant until the end of 8 h. The trend was observed in all four bacterial strains. However, in the presence of Sample 2 different growth patterns could be observed for the different bacterial strains. In the case of Gram-positive *S. aureus*, the number of bacteria decreased gradually from 0–1 h, but after

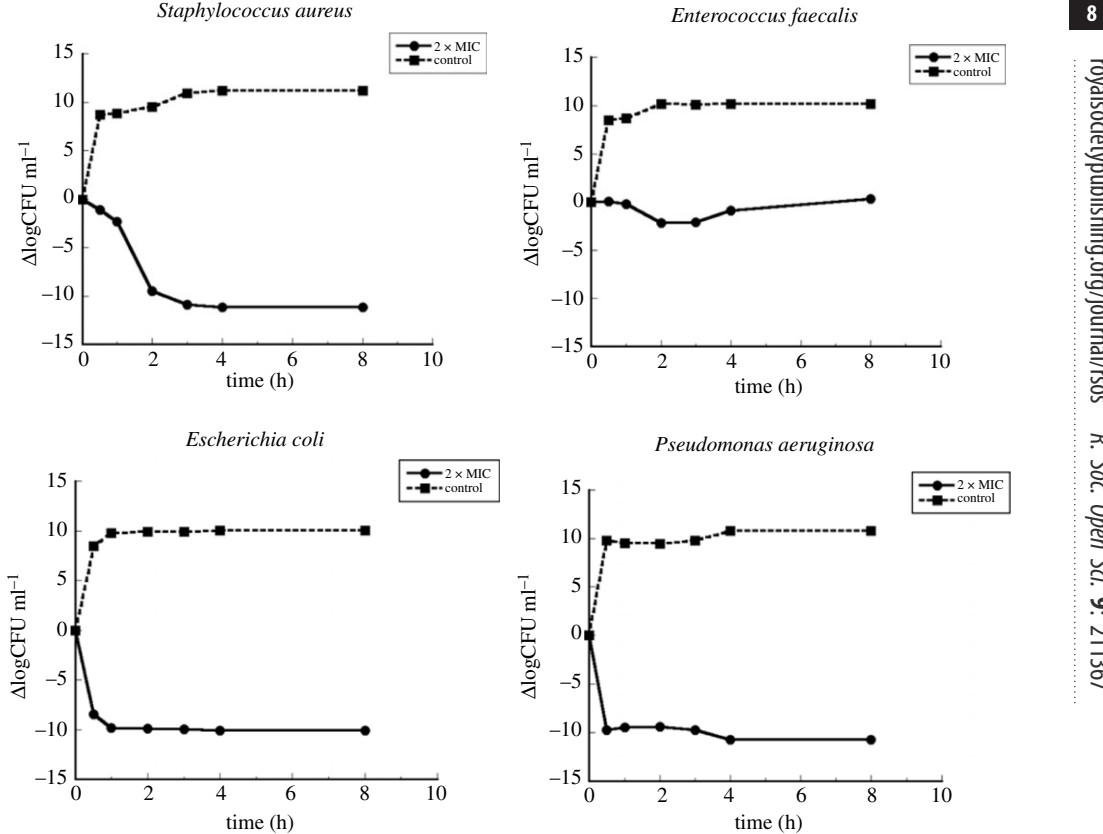

**Figure 2.** Kill curves for Sample 2 against Gram-positive bacteria *S. aureus* and *E. faecalis* and Gram-negative bacteria *E. coli* and *P. aeruginosa*.

**Table 5.** Table showing the MIC and MLC values for the polymers.

| | S. aureus | | E. faecalis | | E. coli | | P. aeruginosa | |
|---|---|---|---|---|---|---|---|---|
| sample | MIC ($\mu g\ cm^{-3}$) | MLC ($\mu g\ cm^{-3}$) | MIC ($\mu g\ cm^{-3}$) | MLC ($\mu g\ cm^{-3}$) | MIC ($\mu g\ cm^{-3}$) | MLC ($\mu g\ cm^{-3}$) | MIC ($\mu g\ cm^{-3}$) | MLC ($\mu g\ cm^{-3}$) |
| 1 | 512 | 1024 | 512 | 512 | 256 | 256 | 2048 | 2048 |
| 2 | 512 | 2048 | 512 | 1024 | 256 | 256 | 512 | 512 |
| 3 | 512 | 1024 | 1024 | 1024 | 256 | 256 | 4096 | 4096 |
| 4 | 512 | 2048 | 512 | 1024 | 128 | 128 | 1024 | 1024 |
| 5 | 512 | 1024 | 512 | 512 | 256 | 256 | 4096 | 4096 |

1–2 h the reduction was significant and from 2 h onwards the number of bacteria further decreased and became constant until the end of 8 h. The effect of Sample 2 towards the other Gram-positive bacteria *E. faecalis* was seen to be much less. No change in the bacterial count was observed during the first hour. During 2–4 h, a slight reduction in the number of bacteria appears, but this difference cannot be seen from 4 h onwards. Towards the two Gram-negative strains *E. coli* and *P. aeruginosa*, a similar pattern of bacterial reduction can be seen in the graphs. A significant and maximum bactericidal effect could be observed within the first 0.5 h which was maintained until the end of 8 h in both strains. Thus, we see that Sample 3 was capable of killing *E. coli* and *P. aeruginosa* to a greater extent within a short interval of time, while the same effect towards *S. aureus* was achieved only gradually with time. Towards *E. faecalis*, no bactericidal effect could be observed within the measured time interval. Hence, we can conclude that Sample 2 was more effective against the Gram-negative strains, compared to the Gram-positive strains.

Antibacterial materials can be used in medicine and pharmacy, e.g. as a substrate for drug carriers, but also to give surfaces antibacterial properties. Obtaining and applying materials with antibacterial properties is important because of the need to maintain hygiene and ensure sterile conditions in

places such as hospitals, the pharmaceutical, food and cosmetology industries. It is particularly important because, as studies show, about 80% of infectious diseases are transmitted by touch.

## 4. Conclusion

The synthesis of bioactive polymer poly(*N*,*N*-dimethylaminoethyl methacrylate) was performed during the research, using different amounts of initiator to obtain a linear product with variable degree of polymerization. Based on viscosity and chromatographic measurements, no clear relationship was found between the obtained results of the average molecular weight and the conditions for performing the polymerization reaction, which suggests that a polymer of branched configuration was most likely obtained. A biocidal impact was shown of all polymers on the Gram-positive (*S. aureus, E. faecalis*) and two Gram-negative (*E. coli and P. aeruginosa*) bacteria. The above observations confirm that PDMAEMA, regardless of its structure, features biocidal properties against a wide range of bacteria.

Data accessibility. The datasets supporting this article have been uploaded as part of the electronic supplementary material [38].

Authors' contributions. D.S.: conceptualization, methodology, resources, supervision, writing-original draft, writing-review and editing; K.R.: data curation, investigation, methodology; D.Z.: data curation, formal analysis, methodology, resources; P.S.: data curation, formal analysis, investigation, methodology; M.M.: conceptualization, formal analysis, methodology, supervision, writing-original draft; M.A.H.: formal analysis, methodology, supervision.

All authors gave final approval for publication and agreed to be held accountable for the work performed therein.

Competing interests. We have no competing interests.

Funding. The manuscript was financed from funds assigned for 14-148-1-21-28 statuary activity, by the Lodz University of Technology, Institute of Material Technologies of Textiles and Polymer Composites, Poland.

Acknowledgments. The authors disclose the receipt of financial support for the research, authorship and/or publication of this article.

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
