## [Peer Review File · Royal Society Open Science]

Review History

RSOS-211367.R0 (Original submission)

Review form: Reviewer 1

Is the manuscript scientifically sound in its present form?

Yes

Are the interpretations and conclusions justified by the results?

Yes

Is the language acceptable?

No

Do you have any ethical concerns with this paper?

Yes

Have you any concerns about statistical analyses in this paper?

Yes

Recommendation?

Major revision is needed (please make suggestions in comments)

Comments to the Author(s)

Please find the attachment (see Appendix A).

Review form: Reviewer 2

Is the manuscript scientifically sound in its present form?

Yes

Are the interpretations and conclusions justified by the results?

No

Is the language acceptable?

No

Do you have any ethical concerns with this paper?

No

Have you any concerns about statistical analyses in this paper?

No

Recommendation?

Major revision is needed (please make suggestions in comments)

Comments to the Author(s)

This study provides information on the synthesis of the compounds as well on the antibacterial activities.

However, the manuscript is difficult to read.

I suggest the author revise and run the manuscript through the grammarly program.

Also, I suggest the author defines where this polymer could be useful for application. This may not be useful for medical purposes as the concentration required is very high for it to be effective.

Also, the introduction section needs major improvement. The thoughts were not connected in the section. Please include chemical structure of the compound for easiest to understand.

Decision letter (RSOS-211367.R0)

Dear Professor Stawski:

Title: Antibacterial properties of poly(N,N-dimethylaminoethyl methacrylate) obtained at different conditions in solution polymerization

Manuscript ID: RSOS-211367

The editor assigned to your manuscript has now received comments from reviewers. We would like you to revise your paper in accordance with the referee and Subject Editor suggestions which can be found below (not including confidential reports to the Editor). Please note this decision does not guarantee eventual acceptance.

Please submit your revised paper before 05-Nov-2021. Please note that the revision deadline will expire at 00.00am on this date. If we do not hear from you within this time then it will be assumed that the paper has been withdrawn. In exceptional circumstances, extensions may be possible if agreed with the Editorial Office in advance. We do not allow multiple rounds of revision so we urge you to make every effort to fully address all of the comments at this stage. If deemed necessary by the Editors, your manuscript will be sent back to one or more of the original reviewers for assessment. If the original reviewers are not available we may invite new reviewers.

Yours sincerely,
Dr Ellis Wilde
Publishing Editor, Journals

On behalf of the Subject Editor Professor Anthony Stace and the Associate Editor Professor Kim Jelfs.

RSC Associate Editor
Comments to the Author:
(There are no comments.)

RSC Subject Editor
Comments to the Author:

(There are no comments.)

Reviewers' Comments to Author:

Reviewer: 1

Comments to the Author(s)

Please find the attachement.

Reviewer: 2

Comments to the Author(s)

This study provides information on the synthesis of the compounds as well on the antibacterial activities.

However, the manuscript is difficult to read.

I suggest the author revise and run the manuscript through the grammarly program.

Also, i suggest the author defines where this polymer could be useful for application. This may not be useful for medical purposes as the concentration required is very high for it to be effective.

Also, the introduction section needs major improvement. The thoughts were not connected in the section. Please include chemical structure of the compound for easiest to understand.

Author's Response to Decision Letter for (RSOS-211367.R0)

See Appendix B.

RSOS-211367.R1 (Revision)

Review form: Reviewer 1

Is the manuscript scientifically sound in its present form?

Yes

Are the interpretations and conclusions justified by the results?

Yes

Is the language acceptable?

Yes

Do you have any ethical concerns with this paper?

No

Have you any concerns about statistical analyses in this paper?

No

Recommendation?

Accept with minor revision (please list in comments)

Comments to the Author(s)

The manuscript after revision is much more improved and suitable for publication. The Authors took all reviewer's comments into consideration. However I have few minor remarks:

1. Title of the manuscript: "Antibacterial properties of poly(N,N-dimethylaminoethylmethacrylate) obtained at different initiator quantities in solution polymerization."
"initiator quantities" is not very suitable for the title, with different amounts of initiator or different concentrations of initiator will be better
2. Materials: p-Xylene for synthesis (Sigma-Aldrich, Germany) was used without any other purification.
Better than "any other" will be "any further".
This term, which also applies to all other reagents, should be changed.

Decision letter (RSOS-211367.R1)

Dear Professor Stawski:

Title: Antibacterial properties of poly(N,N-dimethylaminoethyl methacrylate) obtained at different initiator quantities in solution polymerization
Manuscript ID: RSOS-211367.R1

Thank you for submitting the above manuscript to Royal Society Open Science. On behalf of the Editors and the Royal Society of Chemistry, I am pleased to inform you that your manuscript will be accepted for publication in Royal Society Open Science subject to minor revision in accordance with the referee suggestions. Please find the reviewers' comments at the end of this email.

The reviewers and handling editors have recommended publication, but also suggest some minor revisions to your manuscript. Therefore, I invite you to respond to the comments and revise your manuscript.

Please also include the following statements alongside the other end statements. As we cannot publish your manuscript without these end statements included, if you feel that a given heading is not relevant to your paper, please nevertheless include the heading and explicitly state that it is not relevant to your work. We have included a screenshot example of the end statements for reference.

- Ethics statement

Please clarify whether you received ethical approval from a local ethics committee to carry out your study. If so please include details of this, including the name of the committee that gave consent in a Research Ethics section after your main text. Please also clarify whether you received informed consent for the participants to participate in the study and state this in your Research Ethics section.

OR

Please clarify whether you obtained the necessary licences and approvals from your institutional animal ethics committee before conducting your research. Please provide details of these licences and approvals in an Animal Ethics section after your main text.

OR

Please clarify whether you obtained the appropriate permissions and licences to conduct the fieldwork detailed in your study. Please provide details of these in your methods section.

- Data accessibility

It is a condition of publication that you make available the data and research materials supporting the results in the article. Datasets should be deposited in an appropriate publicly available repository and details of the associated accession number, link or DOI to the datasets must be included in the Data Accessibility section of the article (<https://royalsocietypublishing.org/rsos/for-authors#question17>). Reference(s) to datasets should also be included in the reference list of the article with DOIs (where available).

Please include a Data Availability section after your main text stating where supporting data are available from, or where they will be made available should your article be accepted for publication.

If you wish to submit your supporting data or code to Dryad (<http://datadryad.org/>), or modify your current submission to dryad, please use the following link:
<http://datadryad.org/submit?journalID=RSOS&manu=RSOS-211367.R1>

- Competing interests

Please include a Competing Interests section after your main text declaring any financial or non-financial competing interests. If you have no competing interests please state 'I/we have no competing interests.

- Authors' contributions

Please include an Authors' Contributions section at the end of your main text detailing the contribution of each author. All authors should have read and approved the manuscript before submission and this should be stated in the Authors' Contributions section.

The list of Authors should meet all of the following criteria; 1) substantial contributions to conception and design, or acquisition of data, or analysis and interpretation of data; 2) drafting the article or revising it critically for important intellectual content; and 3) final approval of the version to be published.

- Acknowledgements

- Funding statement

Please include a funding section after your main text which lists the source of funding for each author.

Because the schedule for publication is very tight, it is a condition of publication that you submit the revised version of your manuscript before 26-Nov-2021. Please note that the revision deadline will expire at 00.00am on this date. If you do not think you will be able to meet this date please let me know immediately.

Kind regards,
Dr Ellis Wilde
Publishing Editor, Journals

On behalf of the Subject Editor Professor Anthony Stace and the Associate Editor Professor Kim Jelfs.

RSC Associate Editor
Comments to the Author:
Please make the final changes requested.

RSC Subject Editor
Comments to the Author:
(There are no comments.)

Reviewer comments to Author:
Reviewer: 1
Comments to the Author(s)
The manuscript after revision is much more improved and suitable for publication. The Authors took all reviewer's comments into consideration. However I have few minor remarks:
1. Title of the manuscript: "Antibacterial properties of poly(N,Ndimethylaminoethylmethacrylate) obtained at different initiator quantities in solution polymerization."
"initiator quantities" is not very suitable for the title, with different amounts of initiator or different concentrations of initiator will be better
2. Materials: p-Xylene for synthesis (Sigma-Aldrich, Germany) was used without any other purification.
Better than "any other" will be "any further".
This term, which also applies to all other reagents, should be changed.

Author's Response to Decision Letter for (RSOS-211367.R1)

See Appendix C.

Decision letter (RSOS-211367.R2)

Dear Professor Stawski:

Title: Antibacterial properties of poly(N,N-dimethylaminoethyl methacrylate) obtained at different initiator concentrations in solution polymerization
Manuscript ID: RSOS-211367.R2

It is a pleasure to accept your manuscript in its current form for publication in Royal Society Open Science. The chemistry content of Royal Society Open Science is published in collaboration with the Royal Society of Chemistry.

Yours sincerely,
Dr Ellis Wilde
Publishing Editor, Journals

RSC Associate Editor
Comments to the Author:
(There are no comments.)

Reviewer(s)' Comments to Author:

Appendix A

The manuscript presented to me for review is interesting. The authors investigated the effect of AIBN initiator concentration on the molecular weight of the PDMAEMA polymer and its antibacterial activity against Gram + and Gram - bacteria. They synthesized five polymers with different molecular weight in solvent polymerization. In next step they used several techniques to characterize obtained materials. Most of the conclusions are supported by the presented data. However, there are some questions that should be addressed carefully. Therefore, a major revision is suggested.

Special comments:

1. The title of the manuscript.

Antibacterial properties of poly(N,N-dimethylaminoethylmethacrylate) obtained at different conditions in solution polymerization.

Different suggests that during the reaction different conditions, as temperature, solvent, solvent concentration, etc. are used during the reaction. The authors used different initiator concentrations during studies, so I think that it should be emphasized in the manuscript title.

2. Point 3.1. Material.

First: it should be materials, because authors used more than one compound in their work.

The authors have to give more information about reagents used during the synthesis (such as supplier, purity, and so on).

Moreover, the synthesis of polymers, which is too poorly described, should be more precisely described in the next section: methods.

The monomer name (dimethylaminoethylmethacrylate) should be specified correctly 2(dimethylamino)ethyl methacrylate.

3. Point 3.2. FTIR spectroscopy

The FTIR method should be described in more detail, the measurement parameters should be given. How did the authors prepare the sample to measure by KBr pellets?

4. Point 3.4 Determination of Minimum Inhibitory Concentration (MIC) and Minimum Lethal Concentration (MLC). Page 5 line 11-12

“The samples were prepared by dissolving chitosan derivatives in sterile water to an initial concentration of 32.768 $\mu\text{g}/\text{cm}^3$.”

Something is wrong here, the authors did not investigate the chitosan samples.

5. Point 4. Results and discussion

Page 5 line 60.

“ for Sample 2 should be considered as grossly flawed.”

Was the synthesis repeated? Or did the authors rely on a single synthesis using this (for sample 2) initiator concentration?

Page 6 line 8

“The peak between 2700–300 cm^{-1} is connected with the C-H bond from $-\text{N}(\text{CH}_3)_2$ groups”
It should be 3000 not 300 cm^{-1} .

Page 6 line 43.

Why sample 2, which was name by authors as grossly flawed, was used in this study?

Above, the authors wrote that: The result for Sample 2 should be considered as grossly flawed and as such it will not be taken into account for further consideration. But they used this sample for further research and consideration of obtained results. Moreover, they are particularly focused on this sample when discussing the results. Can authors explain that?

6. Conclusions

Page 7 line 32-33: “The synthesis of bioactive polymer poly(N,N-dimethylaminoethyl methacrylate) was performed during the research, using different amounts of initiator in order to obtain a linear product with a controlled variable degree of polymerisation.”

Does the method of polymer synthesis used allow to obtain a polymer with a controlled mass? For polymer mass control, control radical polymerization are used, such as RAFT, ATRP or NMP. Please explain.

Politechnika Łódzka

Katedra Materialoznawstwa,
Towaroznawstwa i Metrologii Włókienniczej

20th October, 2021

Dear Editors,

thank you very much for Your and Reviewers work connected with our manuscript. We introduced all suggestions made by Reviewers. Our response to each Reviewer is in separate file. Manuscript modified in "Track Changes" function looks very complicated (Article AS2), so for better visibility I prepared also, additionally a pdf copy with all changes accepted (Article AS3). According to English correction we would like to order such possibility at MDPI Editorial Office. I hope Reviewers will be satisfied with current version of the manuscript.

Thank you again for your support
Sincerely Yours

Dawid Stawski, PhD, DSc
Lodz University of Technology
Institute of Material Science
Division of Physical Chemistry of Polymers
Zeromskiego 116, 90-924 Lodz, Poland
phone: +48426313356; mobile: +48500576972,

Politechnika Łódzka

Katedra Materialoznawstwa,
Towaroznawstwa i Metrologii Włókienniczej

20th November, 2021

Dear Reviewer,

thank you very much for your work connected with our manuscript and your valuable input into its content. We introduced your final suggestions and we changed our paper according to them. Below you can see detailed description of the changed made in the paper:

1. *Title of the manuscript: "Antibacterial properties of poly(N,Ndimethylaminoethylmethacrylate) obtained at different initiator quantities in solution polymerization."*

"initiator quantities" is not very suitable for the title, with different amounts of initiator or different concentrations of initiator will be better.

The manuscript title is changed.

2. *Materials: p-Xylene for synthesis (Sigma-Aldrich, Germany) was used without any other purification.*

Better than "any other" will be "any further".

This term, which also applies to all other reagents, should be changed.

We changed "any other" to "any further" in description of all reagents.

Thank you again for your support and I hope that changes which we made are satisfactory for you.

Sincerely Yours

Dawid Stawski, PhD, DSc

Lodz University of Technology

Institute of Material Science

Division of Physical Chemistry of Polymers

Zeromskiego 116, 90-924 Lodz, Poland

phone: +48426313356; mobile: +48500576972,